# Spontaneous Polycystic Kidneys with Chronic Renal Failure in an Aged House Musk Shrew (*Suncus murinus*)

**DOI:** 10.3390/vetsci9030123

**Published:** 2022-03-08

**Authors:** Tohru Kimura

**Affiliations:** Laboratory Animal Science, Joint Faculty of Veterinary Medicine, Yamaguchi University, 1677-1, Yoshida, Yamaguchi 753-8515, Japan; kimura1@yamaguchi-u.ac.jp; Tel.: +81-83-933-5877

**Keywords:** dilatation, house musk shrew (*Suncus murinus*), histopathology, inflammation, interstitial fibrosis, macrophages, polycystic kidneys, renal tubes, serum amyloid A, serum biochemistry

## Abstract

Polycystic kidney disease is one of the most common inheritable renal diseases, characterized by the formation of multiple fluid-filled renal cysts. This disease is a progressive and unfortunately incurable condition. A case of polycystic kidney with chronic renal failure in house musk shrew (*Suncus murinus*) is described. At clinical presentation, a 16-month-old *Suncus murinus* showed weight loss and coarse fur. Regarding the biochemical profile, total protein concentrations increased, resulting in a declined albumin: globulin ratio. Blood urea nitrogen and creatinine concentrations were markedly elevated, indicating the end stage of chronic renal failure. Serum amyloid A levels increased and revealed inflammatory reaction during the cyst formation. Histopathologically, multiple cysts were lined by a single layer of epithelial cells or low cuboidal epithelium. The contents were homogenous eosinophilic materials (mucopolysaccharides or mucoproteins) and these cysts contained abundant macrophages. There were also regeneration and dilatation of renal tubes and interstitial fibrosis. The atrophic glomeruli and glomerular capsules were thickened and hyalinized by dense amorphous mucopolysaccharides. These histopathological findings suggested that the pathogenesis of polycystic kidney disease shared a common mechanistic feature across species.

## 1. Introduction

Polycystic kidney disease is one of the most common inheritable renal diseases, characterized by the formation of multiple fluid-filled renal cysts. This disease affects more than 12 million people in the world with an incident estimated at 1/400 to 1/1000 births [1]. Polycystic kidney disease is a progressive and unfortunately incurable condition that can lead to critical morbidity and renal failure. This lesion causes the progressive formation of cysts leading to eventually the end stage of renal failure. Most therapies for polycystic kidney disease have been taken to address the problem of limiting disease complications mainly renal hypertension [1,2,3].

In veterinary science, congenital polycystic kidneys occur sporadically in many species [4,5,6,7,8,9] but can be inherited as an autosomal dominant lesion in pigs and lambs and can be inherited along with cystic biliary disease in cairn terriers and West Highland white terriers [10,11]. The lesion is inherited as an autosomal dominant trait in families of Persian cats [12,13]. In veterinary medicine, it is proposed that the most expedient way should be to eliminate this disease from future generations of the breeding group.

In contrast, a large number of in vivo models are available for the study of autosomal dominant polycystic kidney-specific disease mechanisms and drug testing. As experimental models, rodent models such as spontaneous mutants and transgenesis present several advantages for biomedical research [14].

The author found spontaneous polycystic kidneys in a laboratory house musk shrew (*Suncus murinus*, family Soricidae, order Insectivore) of a breeding group. *Suncus murinus* is mainly insectivorous and this species feeds on a wide range of invertebrates. It is generally solitary and has a high metabolic rate necessitating frequent feeding.

In the present case report, clinical findings, serum biochemical panel, hematological and inflammatory parameters, and histopathological results of a house musk shrew with polycystic kidney disease associated to chronic renal failure are described.

## 2. Materials and Methods

Male and female *Katmandu* strain *Suncus murinus* served as an outbred stock, which is a breeding group of genetically heterogeneous animals maintained as a closed colony without the introduction of animals from another stock. In this stock animals, we have incidentally encountered a case of cystic kidneys associated to the development of chronic renal failure.

This *Suncus murinus* colony was primarily introduced at 9 weeks of age from Okayama University of Science. The animals were individually housed in cages (CL-0143 (R-2), W 355 × D 499 × H 198 mm, Crea Japan Inc., Tokyo, Japan) kept at a room temperature of 24 ± 2 °C, a relative humidity of 50 ± 5% and an air exchange rate of 15 times/hour. The room was artificially illuminated by 12-h light (07:00–19:00), 12-h dark (19:00–07:00 cycle. The animals had free access to water bottles and a solid diet (d3, Feed One Co., Ltd., Yokahama, Japan).

Under general anesthesia inhalation of 3.5% isoflurane blood samples were collected from the caudal vena cava of *Suncus murinus* using no anticoagulant. At 30 min after collection of blood samples, sera were separated by centrifugation at 1500× *g* for 10 min for biochemical analysis. For hematological samples, blood was collected into tubes containing K_2_EDTA.

In serum biochemical examinations, the following parameters were measured using a blood chemistry analyzer (Dry Chem NX 500V: Fuji Film Co., Ltd., Tokyo, Japan): total protein (TP), albumin (ALB), albumin : globulin (A/G) ratio, total bilirubin (T-BIL), blood urea nitrogen (BUN), creatinine (CRE), urate (UA), glucose (GLU), total cholesterol (TCHO), triglycerides (TG), asparate aminotransferase (AST), alanine aminotransferase (ALT), γ-glutamyl transpeptidase (GGT), lactate dehydrogenase (LDH), alkaline phosphatase (ALP), cholinesterase (ChE), leucine aminopeptidase (LAP), amylase (AMS), creatine kinase (CK), electrolytes (Na, K, Cl, Ca), inorganic phosphorus (IP) and magnesium (Mg).

The following parameters were examined using an automated cell counter (Microsemi LC-662 Horiba Co. Ltd., Kyoto, Japan): white blood cell count (WBC), red blood cell count (RBC), hemoglobin concentration (Hb), packed cell volume (PCV) ratio, mean corpuscular volume (MCV), mean corpuscular hemoglobin (MCH) and mean corpuscular hemoglobin concentration (MCHC).

In acute phase proteins, concentrations of serum amyloid A (SAA) proteins were determined by using a latex agglutination turbidimetric immunoassay (SAA for animals, Eiken Chemical Co., Ltd., Tokyo, Japan, a trial reagent) and autochemistry analyzer method (HITACHI 7170S, Hitachi High-technologies Co., Ltd., Tokyo, Japan).

Immediately after euthanasia, *Suncus murinus* was necropsied and tissue samples were taken for the histopathological examinations. The tissue specimens were fixed in 10% neutral buffered formalin, and 4 μm paraffin sections were stained with hematoxylin and eosine (HE) and periodic acid/Schiff reaction (PAS).

All procedures involving animals were approved by the Institutional Animal Care and Use Committee of Yamaguchi University and followed the Guidelines of Animal Care and Experiments of Yamaguchi University (approval No. 459). The animal care and use program for Advanced Research Center for Laboratory Animal Science in Yamaguchi University has been accredited by AAALAC International (The Association for Assessment and Accreditation of Laboratory Animal Care International) since 2018.

## 3. Results

In the outbred colony, the author found a female, 16-month-old suncus suffering from wasting symptoms such as weight loss (36.1 g; weight range of healthy *Suncus murinus* is 50–70 g for males and 30–50 g for females), malnutrition and coarse fur. The outbred colony of *Suncus murinus* is shown in Figure 1 and physiological and metabolic data in *Suncus murinus* is shown in Table 1.

In this *Suncus murinus* with polycystic kidney disease, cyst formation was not found in other organs including liver and pancreas. Serum biochemical findings examined in this affected animal are shown in Table 2. Although ALB concentrations remained unchanged, TP concentrations appreciably increased, resulting in a declined A/G ratio. Both of the BUN and CRE concentrations were markedly elevated even in the light of the high reference values in BUN for this species.

GLU concentrations dramatically declined from the normal levels to the critical conditions. In contrast, TG levels were moderately raised, and then T-CHO levels greatly increased about four times as much as its reference levels.

Hepatic functional profiles (AST, ALT, GGT, LDH, ChE and LAP) retained the activities of their reference range. In other enzyme activities, AMY and CK indicated increased activities, while a low ALP activity was observed.

In electrolytes, although K and Cl levels considerably increased, Ca and IP concentrations remained stable. SAA used as specific acute inflammatory marker was noticeably high over the reference range.

Hematological results were as follows: WBC (3.80 × 10^9^/L), RBC (6.31 × 10^12^/L), Hb (105 g/L), PCV ratio (0.305), MCV (48.3 fL), MCH (16.6 pg) and MCHC (34.4 g/L); reference values in the laboratory of Laboratory Animal Science: WBC (6.7 ± 2.0 × 10^9^/L), RBC (7.45 ± 0.42 × 10^12^/L), Hb (151.7 ± 7.6 g/L), PCV ratio (0.443 ± 0.010), MCV (59.6 ± 1.6 fL), MCH (20.4 ± 1.2 pg) and MCHC (34.3 ± 1.2 g/dL).

Although not shown in figures, gross appearance of the kidneys was bilaterally involved in numerous enlarged parenchymal cysts. Although the size of the kidney remained unchanged, the surface of the lesions felt rough and irregular. Numerous cysts distended in various sizes were found in the cortex and the medulla. The cyst walls were thin, clear and transparent and the spherical cysts were filled with clear and watery content.

Histopathological findings are shown in Figure 2, Figure 3, Figure 4 and Figure 5. Cyst formation occurred in both of the renal cortex and medulla. These cysts originated not only from the collecting ducts but also from the proximal and distal tubules. Renal specimens showed multiple cystic structures containing homogenous eosinophilic materials (Figure 2). The pinkish materials were concurrent with a large number of macrophages and vacuoles associated with absorption of content (Figure 3). Although few macrophages were found around the injured glomeruli, macrophage infiltration was mainly observed in the cysts, suggesting some associations with the cyst formation and inflammatory reaction. The contents of the cysts were well stained with PAS stain, showing inspissated mucopolysaccharides or mucoproteins (Figure 4). The cysts were lined by a single layer of epithelial cells. Some cysts were lined by flattened to low cuboidal epithelium. The other histological changes were regeneration and dilatation of renal tubes, interstitial fibrosis and atrophy of the glomeruli. The glomeruli and glomerular capsules were thickened and hyalinized by dense amorphous substances, as well as thickening of basement membrane (Figure 5). Interstitial fibrosis was in the areas adjacent to renal cysts and deposition of PAS-positive materials were found in the renal tubules, glomeruli and glomerular capsules.

## 4. Discussion

To the best of the author’s knowledge, this is the first report of polycystic kidney disease with chronic renal failure in a laboratory *Suncus murinus*. The cause of the polycystic kidneys of this *Suncus murinus* could not be examined for its gene expression in the same way as murine gene analysis. However, this renal lesion was likely considered to be acquired polycystic kidney disease in view of the onset of symptoms.

Increased TP and stable ALB concentrations and decreased A/G ratio indicated chronic inflammation and glomerular disease with increase in globulins. As shown in Table 2, *Suncus murinus* are originally characterized by high reference values for BUN, in comparison with those in other laboratory and companion animals [15]. It is probably quite a burden on their renal function during the lifetime. Nevertheless, remarkably increased BUN (renal azotemia) and CRE concentrations in this case revealed the critical features of chronic renal dysfunctions. These findings were in agreement with those described in previous reports in feline patients with polycystic kidney disease [5,7,8,9].

TG and T-CHO concentrations in healthy *Suncus murinus* are considerably low [16,17,18,19,20,21,22]. As shown for the low GLU levels, in the present case the increase in TG and T-CHO concentrations could be due to uremia, sepsis or mobilization of tissue fat stores for energy purpose [23]. Since microcytic and normochromic anemia was mild it is unlikely that it was directly related to uremia or sepsis in this animal.

There were no marked changes in hepatic enzyme activities in this *Suncus murinus*. These results were different from those reported in cats [11] and pigs [9] affected with polycystic kidney disease. This difference is attributable to extrakidney manifestations (polycystic liver disease) observed in cats and pigs. Increases in AMY and CK activities can be attributed to renal failure and muscle damage suffering from wasting disease.

Except for Na and K kevels, marked changes in electrolytes were not found in this suncus, although the other investigation reported severe increases in electrolytes [24]. The influence of electrolytes was probably dependent on the varying stage of renal failure in animals with polycystic kidney disease. The elevated SAA level demonstrated renal inflammation during the cyst formation.

The enlargement and proliferation of cysts caused the compression and obstruction of surrounding nephrons, leading to a decrease in renal function. Cyst formation is always accompanied by extracellular matrix such as proteoglycan, collagen, and elastin [25,26,27]. The PAS-stained findings in renal cysts, tubules and glomeruli agreed with these constituents of extracellular matrix.

Macrophage accumulation was found in the cystic kidney tissues in this *Suncus murinus*, while diffuse mononuclear cell infiltration is most frequently observed in the lesions in chronic kidney diseases alone [28]. Renal macrophages play an important role in homeostasis, surveillance, immune response, tissue injury and repair. Two distinct developmental origins of macrophages are known to be present in the kidneys of laboratory mice. First, infiltrating macrophages are derived from monocyte precursors in the bone marrow and recruited to the kidney in response to local inflammation [29,30]. Secondly, resident macrophages maintain long-term residency in kidney with less mobility and arise primarily during organogenesis. Resident macrophages are generated in the fetal yolk sac, colonize the fetal liver and then migrate into the kidney during early development [31].

Recent important investigation reported that renal macrophages contributed to cystic disease [32,33,34]. Proliferation and accumulation of embryonic-derived resident macrophages trigger renal cyst formation. Injured epithelial cells are struck in a state of persistent injury and these cells are signaling to nearby resident macrophages in an attempt to promote tubule repair. The continued attempt by resident macrophages to repair the epithelium results in enhanced and prolonged epithelial proliferation and renal cyst formation. In the present study, histopathological examinations showed flat epithelial cells and abundant macrophages in numerous polycystic formations. These histopathological findings lend to support to the aforementioned hypothesis in polycystic kidney.

The tissue in polycystic kidney included interstitial infiltration of inflammatory cells and the accumulation of increased macrophages, indicating the interstitial inflammation and the production of chemokines and cytokines [25]. The investigators have described that the interstitial inflammation provokes upregulated expressions of proinflammatory cytokines, activated complement system and altered immune response [25]. The histopathological changes in this study were consistent with an increase in acute phase proteins (SAA). The present results provided additional evidence for inflammation as an important contributor to the cyst growth. Our results suggested that prominent macrophages infiltration and increased SAA levels should be the hallmark of polycystic kidney in this laboratory species animal.

As pointed out in recent studies [25], cyst growth in polycystic kidney disease is associated with fibrosis, inflammation with macrophage aggregation, increases in epithelial cell proliferation, dedifferentiation and fluid secretion. The histopathological results observed in the *Suncus murinus* resembled those of polycystic kidney in humans and rodent models. These findings revealed that the pathogenesis of this disease shared a common mechanistic feature across species including humans.

## 5. Conclusions

In this case report, clinical findings, serum biochemical results, hematological and inflammatory parameters, and histopathological characteristics of polycystic kidney disease in an aged laboratory *Suncus murinus* are described for the first time. The pathophysiological findings in polycystic kidney disease in *Suncus murinus* resembled those observed in humans and animals such as cats, pigs and experimental rodent models. The cyst formation in polycystic kidney disease is associated with inflammation with macrophage, changes in epithelial cell layer, interstitial fibrosis and fluid retention.

## Figures and Tables

**Figure 1 vetsci-09-00123-f001:**
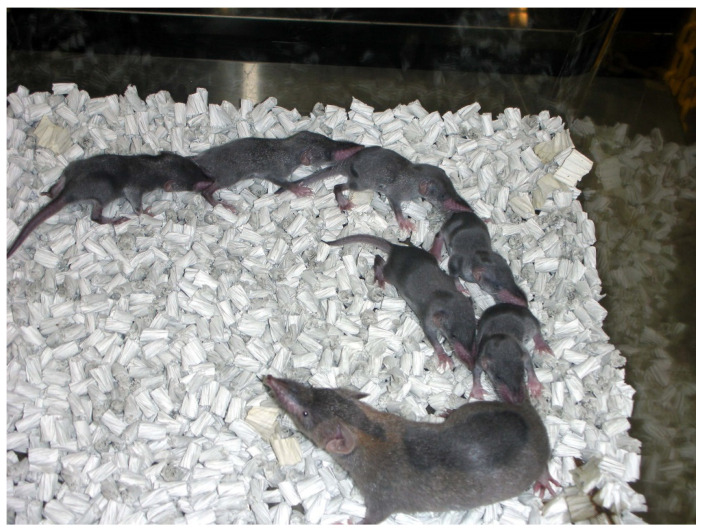
The outbred colony of *Suncus murinus*.

**Figure 2 vetsci-09-00123-f002:**
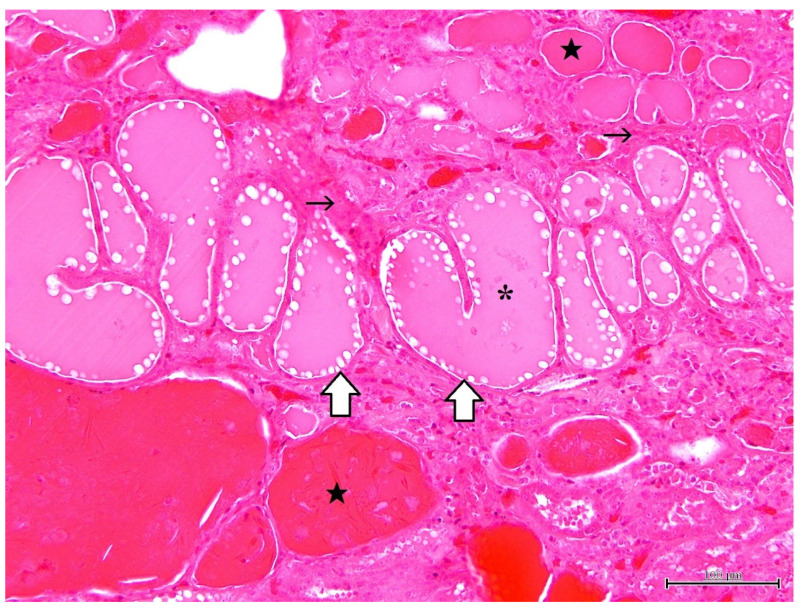
Microscopic finding of polycystic kidney. Note multiple cystic structures (open arrow) containing homogenous pinkish (asterisk) and/or eosinophilic (solid star) materials. Interstitial fibrosis (thin arrow) is also seen. HE stain. ×200, Bar = 100 μm.

**Figure 3 vetsci-09-00123-f003:**
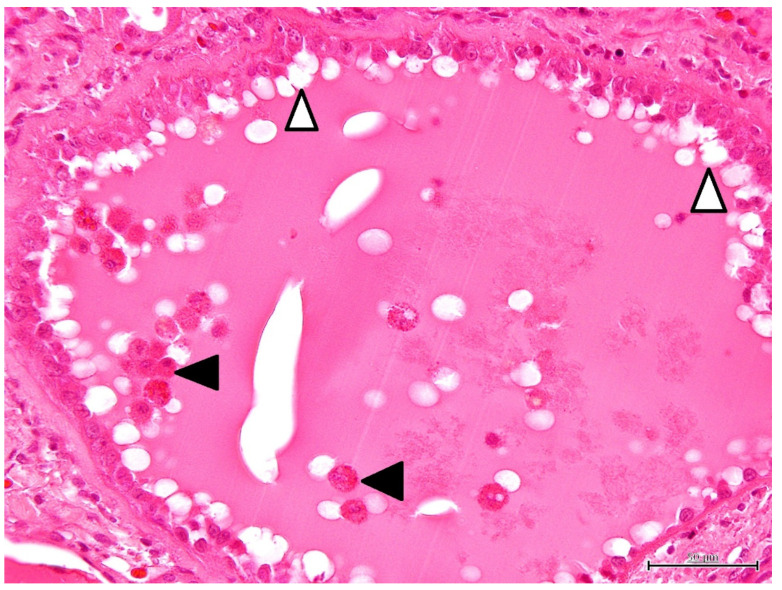
Microscopic finding of polycystic kidney. The cyst contains a large number of macrophages (solid arrowhead). Vacuoles (open arrowhead) is observed around the low cuboidal epithelium. HE stain. ×400, Bar = 50 μm.

**Figure 4 vetsci-09-00123-f004:**
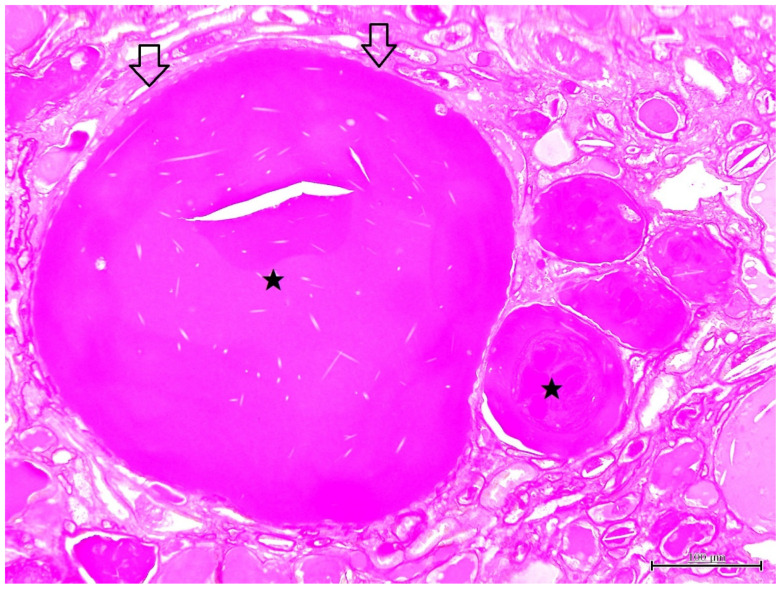
Microscopic finding of polycystic kidney. The cyst contains mucopolysaccharides or mucoproteins (solid star). The cysts are lined by a flattened single epithelium (open arrow). PAS stain. ×200, Bar = 100 μm.

**Figure 5 vetsci-09-00123-f005:**
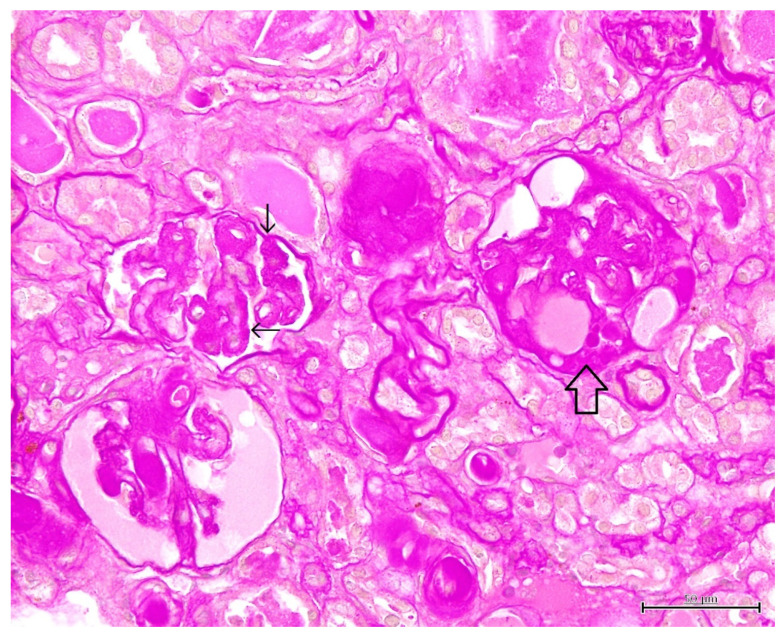
Microscopic finding of polycystic kidney. Dense amorphous substances (open arrow) are deposited in the glomeruli and glomerular capsules. Thickening of basement membrane and hyalinization (thin arrow) are also seen. PAS stain. ×400, Bar = 50 μm.

**Table 1 vetsci-09-00123-t001:** Physiological and metabolic data in *Suncus murinus*.

Items	Data
Chromosome number	40
Body weight at sexual maturity	Male: 50–70 g Female: 30–50 g
The average of life span	1–1.5 years
Sexual cycle	Copulatory ovulator Persistent estrus
The average gestation period	30 days
Range number of offspring (average)	4 to 8 (3)
The range weaning age	20 to 21 days
The average basal metabolic rate	0.403 W

**Table 2 vetsci-09-00123-t002:** Serum biochemical findings in *Suncus murinus* with polycystic kidneys.

Parameters	Measurements	Reference Values (Mean ± SD) *
TP (g/L)	70	54.9 ± 4.2
Alb (g/L)	21	23.6 ± 3.0
A/G ratio	0.43	0.74 ± 0.09
T-BIL (μmol/L)	1.71	3.08 ± 2.05
BUN (mmol/L)	>49.98	23.75 ± 4.36
CRE (μmol/L)	114.39	41.18 ± 25.93
UA (μmol/L)	47.58	111.23 ± 63.64
GLU (mmol/L)	2.94	12.37 ± 5.19
T-CHO (mmol/L)	3.99	0.96 ± 0.19
TG (mmol/L)	0.53	0.28 ± 0.13
AST (U/L)	285	590.87 ± 222.99
ALT (U/L)	78	255.33 ± 146.51
GGT (U/L)	17	24.26 ± 14.87
LDH (U/L)	186	606.87 ± 284.13
ALP (U/L)	1	61.07 ± 21.17
ChE (U/L)	1	1.20 ± 0.78
LAP (U/L)	26	34.93 ± 7.10
AMY (U/L)	2234	1360.87 ± 213.43
CK(U/L)	>2000	1288.73 ± 684.33
Na (mmol/L)	169	161.87 ± 4.76
K (mmol/L)	6	4.43 ± 0.73
Cl (mmol/L)	143	121.60 ± 3.66
Ca (mmol/L)	4.15	5.22 ± 0.33
IP (mmol/L)	1.823	2.94 ± 0.53
Mg (mmol/L)	2.05	1.70 ± 0.21
SAA (μg/mL)	16	1.90 ± 1.19

*: Reference values in the laboratory of Laboratory Animal Science (*n* = 15). TP: total protein, ALB: albumin, A/G: albumin : globulin ratio, T-BIL: total bilirubin, BUN: blood urea nitrogen, CRE: creatinine, UA: urate, GLU: glucose, TCHO: total cholesterol, TG: triglycerides, AST: asparate aminotransferase, ALT: alanine aminotransferase, GGT: γ-glutamyl transpeptidase, LDH: lactate dehydrogenase, ALP: alkaline phosphatase, ChE: cholinesterase, LAP: leucine aminopeptidase, AMS: amylase, CK: creatine kinase, Na: sodium, K: potassium, Cl: chloride, Ca: calcium, IP: inorganic phosphorus and Mg: magnesium.

## Data Availability

Data are available upon reasonable request to the corresponding author.

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
