# Peer review of "Spontaneous Polycystic Kidneys with Chronic Renal Failure in an Aged House Musk Shrew (Suncus murinus)"

_vetsci, 2022, doi:10.3390/vetsci9030123_

Round 1
Reviewer 1 Report
1. The gross appearance of the kidneys was described in lines 114-118. This finding is necessary to diagnose polycystic kidney disease. The macroscopic photographs of the kidneys should be added to the figures.
2. The cause of the polycystic kidneys of this suncus should be discussed. Is it a congenital (polycystic kidney disease: PKD), acquired, or developmental disease?
3. The author suspected that the increase of serum TG and TCho levels might be a cause of hypoglycemia. However, uremia is also known as a cause of hypoglycemia. In addition, sepsis is also known as the major cause of hypoglycemia, and high serum amyloid level was shown in this suncus. Without the result of the complete blood count (CBC), it is difficult to exclude sepsis as the cause of hypoglycemia.
4. The involvement of macrophage infiltration was thoroughly described in the discussion section, but there were no descriptions regarding macrophage infiltration in the result section. However, more high-quality staining or immunohistochemistry would be required to assess the macrophage infiltration.
5. lines 69 and 87; reagents and dose of the anesthesia should be described.
6. TP, A/G, BUN, CRE; abbreviations are not usually be used in the abstract.
7. line 132; amorphous mucopolysaccharides. PAS-positive materials are not always mucopolysaccharides. The “amorphous substances” would be better.
Author Response
Reviewer 1
Open Review
I am grateful to Reviewer #1 for the critical comments and useful suggestions that have helped me to improve our paper. The comments of Reviewer #1 have been helpful in allowing me to revise this manuscript. I have attempted to address the questions raised by Reviewer #1 according to the following.
- The gross appearance of the kidneys was described in lines 114-118. This finding is necessary to diagnose polycystic kidney disease. The macroscopic photographs of the kidneys should be added to the figures.
I regret that I have not taken macroscopic photographs of the lesions. I cannot show the macroscopic photographs in the Figures.
- The cause of the polycystic kidneys of this suncus should be discussed. Is it a congenital (polycystic kidney disease: PKD), acquired, or developmental disease?
Page 7, lines 196-199.
The sentences have been added in the paragraph.
“The cause of the polycystic kidneys of this suncus could not be examined for its gene expression in the same way as murine gene alalysis. However, this renal lesion was likely considered to be acquired polycystic kidney disease in view of the onset of symptoms.”
- The author suspected that the increase of serum TG and TCho levels might be a cause of hypoglycemia. However, uremia is also known as a cause of hypoglycemia. In addition, sepsis is also known as the major cause of hypoglycemia, and high serum amyloid level was shown in this suncus. Without the result of the complete blood count (CBC), it is difficult to exclude sepsis as the cause of hypoglycemia.
I agreed your suggestions.
Page 7, lines 210-212 .
The sentences have been revised as pointed out.
“As shown in the low GLU levels, the present case likely caused an increase in TG and T-CHO concentrations following uremia , sepsis or mobilization of tissue fat stores for energy purpose [23].”
- The involvement of macrophage infiltration was thoroughly described in the discussion section, but there were no descriptions regarding macrophage infiltration in the result section. However, more high-quality staining or immunohistochemistry would be required to assess the macrophage infiltration.
Page 4, lines 160-162.
The sentences have been added in the Results section.
“Although few macrophages were found around the injured glomeruli, macrophage infiltration was mainly observed in the cysts, suggesting some associations with the cyst formation and inflammatory reaction.”
- lines 69 and 87; reagents and dose of the anesthesia should be described.
Page 2, lines 69-72. Page 2, lines 86-89. the sentences have been corrected as suggested.
“Under systemic anesthesia with 3.5% isoflurane, blood samples were collected from the caudal vena cava of the suncus using no anticoagulant.”
“The suncus was euthanized by carbon dioxide exposure (a displacement rate 40% of the chamber volume/min). Immediately after euthanasia with carbon dioxide exposure, the suncus was necropsied and tissue samples were taken for the histopathological examinations.”
- TP, A/G, BUN, CRE; abbreviations are not usually be used in the abstract.
Page 1, lines 13-15. The sentences have been corrected as suggested. Abbreviations → full names.
Serum biochemically, total protein concentrations increased, resulting in a declined albumin: globulin ratio. Blood urea nitrogen and creatinine concentrations were markedly elevated, indicating the end stage of chronic renal failure.
- line 132; amorphous mucopolysaccharides. PAS-positive materials are not always mucopolysaccharides. The “amorphous substances” would be better.
“amorphous mucopolysaccharides”→ “amorphous substances”
Page 5, lines 167-168. Page 7, lines 189-191.
“The glomeruli and glomerular capsules were thickened and hyalinized by dense amorphous substances, as well as thickening of basement membrane (Figure 4).”
“Figure 4. Microscopic finding of polycystic kidney.
Dense amorphous substances are deposited in the glomeruli and glomerular capsules. Thickening of basement membrane and hyalinization are also seen. PAS stain. × 400, Bar = 50 μm.”
Reviewer 2 Report
The case report by Tohru Kimura reveals a new finding of polycystic kidney disease (PKD) with chronic renal failure in a female suncus. The author provided serum biochemical profiles and kidney histopathological evidence suggesting PKD was developed in a female suncus. The author reported that the changes in the suncus kidney including histopathology and inflammatory reactions which were similar to the characteristics of PKD reported in human and other mammal species. This manuscript is potentially important to discover a new case of PKD with kidney dysfunction in a female suncus, but there are some issues that need to be addressed.
Major comments
Since this is the first case report that PKD with kidney failure was found in a female suncus, did the author check the PKD gene expression in this PKD suncus?
Other minor Comments
In Abstract, line 10: do “the contents” mean “the contents of the kidney cysts”? Additionally, it would be great to give definitions for TP and A/G or full names rather than abbreviations in the abstract.
Introduction
- Paragraph 1, Should use "multiple fluid-filled renal cysts" instead of "multiple full-filled renal cysts"?
- Paragraph 1, “(Raptis, Foo, Colbert,)” should delete those authors' names.
Materials and Methods
- The author stated in the title “in an aged suncus” and in the Method section “this suncus colony was primarily introduced at 9 weeks of age from Okayama University of Science”. However, the exact age of this female suncus with PKD was not provided in this case report.
- Be consistent with abbreviations. For example, total bilirubin was defined as TBIL in Methods but T-BIL was used in Table 1.
- In paragraph 5, serum concentrations of serum amyloid A (SAA) proteins. Should the second serum be deleted?
Results
- What was the size of the PKD kidneys compared to normal suncus kidneys?
- In paragraph 2, lines 1-2“cyst formation was not found in different organs including liver and pancreas” instead of different organs should be in other organs.
- How many samples are in your reference values (Table 1)?
Discussion
- In paragraph 2, lines 3-4: it is unclear what this sentence means? Did the author intend to suggest that BUN was higher in the reference values (Table 1) compared to the reports by other laboratories? Is it true, it would be nice to provide citations for this statement?
- Page 7, paragraph 6, "provided evidence additional evidence", this sentence was repeated.
Author Response
Reviewer 2
Open Review
The case report by Tohru Kimura reveals a new finding of polycystic kidney disease (PKD) with chronic renal failure in a female suncus. The author provided serum biochemical profiles and kidney histopathological evidence suggesting PKD was developed in a female suncus. The author reported that the changes in the suncus kidney including histopathology and inflammatory reactions which were similar to the characteristics of PKD reported in human and other mammal species. This manuscript is potentially important to discover a new case of PKD with kidney dysfunction in a female suncus, but there are some issues that need to be addressed.
I am grateful to Reviewer #2 for the critical comments and useful suggestions that have helped me to improve this paper. As indicated in the response that follow, I have taken all these comments and suggestions into account in the revised version of this paper.
Major comments
Since this is the first case report that PKD with kidney failure was found in a female suncus, did the author check the PKD gene expression in this PKD suncus?
Page 7, lines 196-199.
The sentences have been added in the paragraph.
“The cause of the polycystic kidneys of this suncus could not be examined for its gene expression in the same way as murine gene alalysis. However, this renal lesion was likely considered to be acquired polycystic kidney disease in view of the onset of symptoms.”
Other minor Comments
In Abstract, line 10: do “the contents” mean “the contents of the kidney cysts”? Additionally, it would be great to give definitions for TP and A/G or full names rather than abbreviations in the abstract.
Page 1, lines 13-15. The sentences have been corrected as suggested. Abbreviations → full names.
Serum biochemically, total protein concentrations increased, resulting in a declined albumin: globulin ratio. Blood urea nitrogen and creatinine concentrations were markedly elevated, indicating the end stage of chronic renal failure.
Introduction
- Paragraph 1, Should use "multiple fluid-filled renal cysts" instead of "multiple full-filled renal cysts"?
Page 1, line 9. Page 1, line 29.
The expression has been changed as suggested.
multiple filled-filled renal cysts → multiple fluid-filled renal cysts
- Paragraph 1, “(Raptis, Foo, Colbert,)” should delete those authors' names.
Page 1, line 35
I am sorry for my careless mistake.
(Raptis, Foo, Colbert,) has been deleted.
Materials and Methods
- The author stated in the title “in an aged suncus” and in the Method section “this suncus colony was primarily introduced at 9 weeks of age from Okayama University of Science”. However, the exact age of this female suncus with PKD was not provided in this case report.
Page 3, lines 98-100.
The exact age has been added in the sentence.
“In the outbred colony, the author found a female, 16-month-old suncus suffering from wasting symptoms such as weight loss (36.1 g), emaciation, malnutrition and coarse fur.”
- Be consistent with abbreviations. For example, total bilirubin was defined as TBIL in Methods but T-BIL was used in Table 1.
Page 2, line 75.
The abbreviation has been corrected.
“TBIL” → “T-BIL”
- In paragraph 5, serum concentrations of serum amyloid A (SAA) proteins. Should the second serum be deleted?
Page 2, lines 82-85.
The first serum has been deleted.
“In acute phase proteins, concentrations of serum amyloid A (SAA) proteins were determined by using a latex agglutination turbidimetric immunoassay (SAA for animals, Eiken Chemical Co., Ltd., Tokyo, Japan, a trial reagent) and auto-chemistry analyzer method (HITACHI 7170S, Hitachi High-technologies Co., Ltd., Tokyo, Japan).”
Results
- What was the size of the PKD kidneys compared to normal suncus kidneys?
Page 3, lines 117-120.
The sentences have corrected. The size of the PKD kidneys was normal.
“Gross appearance of the kidneys was bilaterally involved in numerous enlarged parenchymal cysts. Although the size of the kidney remained unchanged, the surface of the lesions felt rough and irregular.”
- In paragraph 2, lines 1-2“cyst formation was not found in different organs including liver and pancreas” instead of different organs should be in other organs.
Page 3, lines 101-102.
The sentence has been corrected as suggested.
“In this suncus with polycystic kidney disease, cyst formation was not found in other organs including liver and pancreas.”
- How many samples are in your reference values (Table 1)?
Page 3, lines 102-104. Page 4, lines 153.
The sentence has been corrected. Sample size is 15.
Serum biochemical findings examined in this affected animal are shown in Table 1, in contradistinction to the author’s reference values in healthy suncus (n = 15).
Table 1. footnote has also been corrected.
“*: Reference values in the laboratory of Laboratory Animal Science (n = 15).”
Discussion
- In paragraph 2, lines 3-4: it is unclear what this sentence means? Did the author intend to suggest that BUN was higher in the reference values (Table 1) compared to the reports by other laboratories? Is it true, it would be nice to provide citations for this statement?
Page 7, lines 201-203.
The meaning of the sentence has been clarified as suggested.
As shown in Table 1, suncus are originally characterized by high reference values for BUN, in comparison with those in the other laboratory animals and companion animals [15].
- Page 7, paragraph 6, "provided evidence additional evidence", this sentence was repeated.
Page 8, lines 255-256.
The sentence has been corrected.
“The present results provided additional evidence for inflammation as an important contributor to the formation of cyst growth.”
Round 2
Reviewer 1 Report
Dear Author;
I can accept your revision, although the macroscopic figures are lacking, unfortunately. Please correct the sentence below.
Line 69: 3.5% isoflurane, please change to "inhalation of 3.5% isoflurane".
Author Response
Reviewer #1
Comments and Suggestions for Authors
Dear Author;
I can accept your revision, although the macroscopic figures are lacking, unfortunately. Please correct the sentence below.
Line 69: 3.5% isoflurane, please change to "inhalation of 3.5% isoflurane".
Reviewer #1
Thank you very much for your help. I am most grateful for your kindness.
Page 2, lines 69-70.
The sentence has been corrected as suggested.
“with 3.5% isoflurane” → “inhalation of 3.5% isoflurane”
